# The Tobacco Smoke Component, Acrolein, as a Major Culprit in Lung Diseases and Respiratory Cancers: Molecular Mechanisms of Acrolein Cytotoxic Activity

**DOI:** 10.3390/cells12060879

**Published:** 2023-03-11

**Authors:** Pawel Hikisz, Damian Jacenik

**Affiliations:** 1Department of Oncobiology and Epigenetics, Faculty of Biology and Environmental Protection, University of Lodz, ul. Pomorska 141/143, 90-236 Lodz, Poland; 2Department of Cytobiochemistry, Faculty of Biology and Environmental Protection, University of Lodz, ul. Pomorska 141/143, 90-236 Lodz, Poland; damian.jacenik@biol.uni.lodz.pl

**Keywords:** acrolein, lung cancer, inflammation, tobacco smoke, COPD, respiratory toxicity, oxidative stress, DNA adducts, cell signalling, α,β-unsaturated aldehydes

## Abstract

Acrolein, a highly reactive unsaturated aldehyde, is a ubiquitous environmental pollutant that seriously threatens human health and life. Due to its high reactivity, cytotoxicity and genotoxicity, acrolein is involved in the development of several diseases, including multiple sclerosis, neurodegenerative diseases such as Alzheimer’s disease, cardiovascular and respiratory diseases, diabetes mellitus and even the development of cancer. Traditional tobacco smokers and e-cigarette users are particularly exposed to the harmful effects of acrolein. High concentrations of acrolein have been found in both mainstream and side-stream tobacco smoke. Acrolein is considered one of cigarette smoke’s most toxic and harmful components. Chronic exposure to acrolein through cigarette smoke has been linked to the development of asthma, acute lung injury, chronic obstructive pulmonary disease (COPD) and even respiratory cancers. This review addresses the current state of knowledge on the pathological molecular mechanisms of acrolein in the induction, course and development of lung diseases and cancers in smokers.

## 1. Introduction

Despite the indisputable harmful effect of smoking on general health, condition of the body or fertility in men, smoking is still one of the most commonly used and widely available stimulants. The World Health Organization (WHO) informs that almost one third of the adult population are smokers. These figures include both daily and occasional smokers. Worldwide, smoking prevalence remains high, with an estimated 19% of the adult population using tobacco products: 33% among men and 6% among women [1]. Numerous studies underline the unequivocal causal relationship between tobacco use and morbidity and mortality associated with cardiovascular diseases and cancer. Smoking all tobacco products, including the most popular cigarettes, causes the development of many health and life-threatening disorders, including within the circulatory and respiratory systems, cancer, fertility disorders and fetal malformations. It is also one of the leading causes of the highest number of premature deaths. The main reason is that the chemicals released undergo a series of changes during the smoking process of tobacco products [2,3].

There are more than 9500 highly complex mixtures of substances that act on the body in tobacco and tobacco smoke [4]. Many have harmful, toxic, carcinogenic and teratogenic properties (causing defects in fetal development). Tobacco smoke is harmful to the smoker and to the people around him (so-called passive smoking). Therefore, it is worth knowing what we expose ourselves and others to by smoking cigarettes. Better methods of tobacco ingredient analysis have allowed for more accurate analysis and verification of individual tobacco ingredients. The latest research published by the International Agency for Research on Cancer (IARC) shows that the total number of carcinogens classified in tobacco and tobacco smoke has increased to 83 since 2022 [2].

All components of tobacco smoke undergo subsequent changes during smoking, e.g., oxidation, hydrogenation, pyrolysis or hydrolysis. The main part of the mass of smoke from a cigarette consists of components containing oxygen, carbon dioxide and nitrogen (approx. 90%), while the remaining part (approx. 10%) is miscellaneous components [4]. Regarding the works of Roberts [5] and Hoffmann [6], these compounds, depending on their functional groups, can be divided into hydrocarbons, amines, N-nitrosamines, ethers, aldehydes, halogenated compounds, nitro compounds, phenolic compounds, miscellaneous compounds, and inorganic compounds [2]. A wide range of carcinogens and mutagens have been found in cigarette smoke, including radioactive polonium, benzopyrene, dimethylbenzanthracene, naphthalene, methylnaphthalene, polycyclic aromatic hydrocarbons and heavy metals such as cadmium. The particularly harmful polycyclic aromatic hydrocarbons (PAH), tobacco-specific N-nitrosamines (TSNAs) and aldehydes should be emphasized here [2,7,8].

Aldehydes identified in tobacco smoke include formaldehyde (Group 1), acrolein (Group 2A), acetaldehyde and crotonaldehyde (Group 2B) and are classified by the IARC as carcinogens. Importantly, they usually occur in much higher concentrations than PAHs and TSNA, ranging from μg/cigarette to mg/cigarette. The formation of aldehydes is primarily due to tobacco combustion and pyrolysis, as these compounds are barely detected in unburnt tobacco [9,10]. Unfortunately, as Li et al. [2] point out in the pooled analysis, there is no clear downward trend in the occurrence of these carcinogenic aldehydes in tobacco smoke.

Increasingly, the latest studies indicate that acrolein present in cigarette smoke is particularly harmful to health. While there are many sources of exposure to acrolein, smoking is thought to be the greatest. Moreover, the health impacts from the inhalation of acrolein are more significant than from other routes of exposure [11]. It turns out that this highly reactive α,β-unsaturated aldehyde causes the formation of chronic obstructive pulmonary disease (COPD) in smokers; additionally, thanks to its strong immunomodulatory and genotoxic properties, it induces the formation of cancer, e.g., lungs [12,13,14,15]. Many studies in mouse/rat research models have shown that acrolein is a serious threat to the respiratory system. Its exposure leads to the induction of many diseases, including cancer. A very valuable meta-analysis of the relationship between acrolein exposure and respiratory system damage in rats/mice was made by Liu et al. [16]. As the authors indicate, exposure to acrolein can significantly induce an inflammatory response in the lungs of rats/mice [16].

We present here the main mechanisms of biological activity of acrolein in terms of COPD and lung cancer in smokers, with particular emphasis on the molecular effects of this aldehyde.

## 2. Methodology

A search for relevant literature for this review was conducted on three electronic databases—PubMed, Scopus, and Google Scholar. Each chosen article was published before 1 February 2023 and written in English. The following search terms and logical hyphens were included in the search strategy: “acrolein tobacco smoke”, “acrolein e-cigarettes”, “acrolein toxicity”, “acrolein lungs cancer”, and “acrolein lungs disorders”. These terms were searched for in the abstract, title or keywords. The relevance of the articles was estimated by analyzing their title and abstract. The articles meeting all search criteria were full-text evaluated and included in this review.

## 3. Cigarette Smoke—Health Effects and Risks

Due to its physical and chemical properties, cigarette smoke is an aerosol formed in the incomplete combustion of tobacco. The effect of all harmful substances from tobacco smoke is not limited only to tobacco smokers but also affects all people who stay in rooms where cigarettes are smoked (so-called “passive smoking”) [8,17]. According to numerous studies and recent data, tobacco and tobacco smoke have been found to contain a complex mixture of over 9500 chemical compounds. Many of them have been recognized as dangerous to human health. In 2012, the U.S. Food and Drug Administration compiled a list of harmful and potentially harmful components in unburned tobacco and tobacco smoke, 79 of which are considered carcinogens. Over the last 10 years, thanks to the development of science and modern instrumental and analytical methods, new data are constantly emerging that increase our knowledge about the levels of carcinogens in tobacco products. To date, 83 carcinogens have been identified—37 in unburnt tobacco and 80 in tobacco smoke. Importantly, no clear downward trend has been observed for these carcinogens in recent years [2].

Recent papers by Li and Hecht [2] and Nardone et al. [18] provide a valuable characterization of toxic substances in cigarette smoke. All components of tobacco smoke undergo subsequent changes during smoking, e.g., by oxidation, reduction, pyrolysis, or hydrolysis. Each of these substances often has a complex biological effect that harms smokers. In addition to nicotine, the primary alkaloid responsible for tobacco’s effects and addictive properties, a wide range of carcinogens and mutagens have been detected in cigarette smoke, including carbon monoxide, hydrogen cyanide, sulfur oxide, ammonia, formaldehyde, acrolein, benzene, vinyl chloride, heavy metals such as lead, nickel, toluidine, arsenic, naphthylamine, dimethyl nitrosamine, dibenzo acridine, polycyclic aromatic hydrocarbons (PAHs) [2,18].

There is ample evidence of a correlation between smoking tobacco products and an increase in the incidence of cardiovascular diseases and increased incidence and mortality of various types of cancer [19,20]. As indicated by Hackshaw et al. [21], even light tobacco use (one cigarette a day) increases the risk of developing ischemic heart disease by 48–74%. In the case of smokers who smoke 20 cigarettes a day, ischemic heart disease was observed twice as often [21]. A similar correlation between the dose and the body’s response is observed in the case of cancer incidence. With an increase in the consumption of tobacco products by smokers, there is an increase in the incidence of various types of cancer, such as lung [22], ovarian [23], bladder [24], prostate [25], or pancreatic [26]. Importantly, recent results have also highlighted the adverse effects of cigarette smoke on male fertility through, among other things, the generation of reactive oxygen species and DNA damage resulting in decreased sperm quality and sperm activity [3,27].

## 4. Acrolein and Cigarette Smoke and Smoking

Smoking tobacco products is undoubtedly one of the primary sources of the exposure of the human body to acrolein. As shown, the analysis of the concentration of acrolein metabolites in the urine of smokers indicates elevated levels in smokers and even in passive smokers. Moreover, after four weeks of smoking abstinence, a significant decrease in acrolein metabolites was observed, indicating a reduced exposure of the subjects to the aldehyde [28,29]. Of course, it should be emphasized that acrolein, in its pure chemical form, is not added as an ingredient to cigarettes. Its presence in cigarettes results from the chemical reactions accompanying the burning of a cigarette and the transformation of glycerol and sugars contained in them [30,31]. Thus, smokers inhale acrolein when they inhale the smoke (first-hand smoke). In smaller amounts, it is also inhaled by passive smokers (second-hand smoke) [28].

Glycerin is added to the tobacco in about 1–5% by weight to maintain moisture and absorb added flavors. It has been shown that its presence in the composition of cigarettes increases the concentration of acrolein in cigarette smoke [30]. It is worth noting, however, that the emission of acrolein in the case of cigarettes containing glycerin is only slightly higher compared to non-glycerin cigarettes. A much more important source of acrolein in tobacco products is carbohydrates. Their concentration, depending on the type of tobacco product, is crucial to the concentration of acrolein in cigarette smoke [31]. Unprocessed tobacco leaves contain many sugars, including glucose, fructose and sucrose. Drying (curing) the leaves may affect their concentration; while air-cured tobacco contains virtually no sugars, flue-cured tobacco can contain concentrations up to 25% of its weight. In addition, various sugar components containing honey and fruit syrups during production as a binder, coating component, flavoring, formulation aid or humectant are added directly during the commercial manufacture of cigarettes [11]. Studies indicate a positive correlation between the presence of sugars (including sucrose and fructose) in tobacco products, the increased concentration of acrolein in cigarette smoke, and other harmful aldehydes such as formaldehyde and acetaldehyde. For example, adding 16% sucrose to cigarettes increased from 118 μg to 215 μg of acrolein per cigarette, indicating that carbohydrates are the primary source of acrolein in cigarettes [10]. The latest study by Pennings et al. [9], a multivariate analysis, confirms a solid proportional relationship between the concentration of harmful and carcinogenic aldehydes in cigarette smoke and the concentration of sugars and humectants. The study’s authors emphasize that the emissions of formaldehyde, acetaldehyde, acrolein and 2,5-dimethylfuran in tobacco smoke may decrease when the sugar and humectant content of cigarettes is reduced [9].

Considering these data, it is difficult to indicate what concentrations of acrolein a cigarette smoker is exposed to while smoking. The value of the aldehyde concentration is, of course, conditioned by smoking conditions, puff volume, puff rate, the types and brands of cigarettes, the method of their manufacture and composition, which is clearly shown by research by independent teams [9,32,33] where the concentration of acrolein expressed in µg/cigarette was different. The latest data from 2021 from the International Agency for Research on Cancer [34] indicate that the concentration of acrolein in cigarette smoke of various cigarettes ranges from 24.9 to 223 µg/cigarette. Interestingly, acrolein concentrations in mainstream smoke (defined as smoke exhaled directly by the smoker) appear to be significantly lower than concentrations in side stream smoke (defined as smoke emitted from smoldering tobacco between puffs) due to altered combustion chemistry at lower temperatures [35,36,37]. In smoky rooms, such as bars and restaurants, acrolein levels are between 2.3–275 µg/m^3^ [23].

## 5. Acrolein and E-Cigarettes—A Healthier Alternative?

E-cigarettes are becoming more and more popular among people of all ages. Unfortunately, juveniles are also increasingly using this type of stimulant. It is estimated that in 2020 there were 68 million adult e-cigarette users worldwide [38]. In the United States in 2021, 2.06 million middle and high school students used e-cigarettes in the past 30 days, while in 2019, 4.5–4.8% of adults were current e-cigarette users [39,40]. It is worth noting that the effects of long-term use of e-cigarettes on health are still not fully known and remain under constant discussion. Undoubtedly, however, electronic smoking devices pose a serious threat to global health and are one of the current public health problems; it is growing at an accelerating pace and has become a public health emergency [41,42]. E-cigarette manufacturers convince consumers that electric smoking devices pose a lower health risk by reducing carcinogenic and harmful substances, however, the latest research indicates otherwise [41,42,43,44]. Heating the products in electronic cigarettes produces condensed carcinogenic hydrocarbons and products toxic to the human body. One example is the presence of nickel and ethylene oxide, substances linked to lung and sinus cancers, lymphomas, multiple myeloma and leukemia [42]. As indicated by Lorkiewicz et al. [41], there are known studies on mouse models and humans where the harmful effects of e-cigarettes are indicated [41]. Exposure to e-cigarette smoke causes endothelial dysfunction, pulmonary and cardiovascular injury by raising heart rate and blood pressure, and even DNA mutations leading to cancer [42]. A valuable mini-review from Guo and Hecht [45] provides a good evaluation of the in vitro and in vivo studies on the harmful effects of e-cigarettes. The research summary indicates that e-cigarette liquids or vapor are highly cytotoxic to users and, moreover, capable of inducing DNA damage, oxidative stress, DNA double-stranded breaks, apoptosis, and genotoxicity in different types of oral cells [45]. Undoubtedly, the harmful effects of e-cigarettes are associated, among others, with reactive aldehydes such as formaldehyde or acrolein generated in e-cigarette aerosols. E-cigarette aerosols, like mainstream or side stream cigarette smoke, have been shown to contain measurable amounts of reactive carbonyls such as acrolein, formaldehyde and acetaldehyde [46,47]. As indicated by several recent studies [48,49,50,51], the degree of formation of these aldehydes depends on many factors, such as the operating conditions of the e-cigarette device (e.g., power), the topography of the user and the relative abundance of propylene glycol (PG) and vegetable glycerin (VG) in the e-liquid.

Notably, as outlined in the latest research, several groups indicate that VG or PG thermal degradation during smoking is crucial to producing aldehydes, including acrolein, in e-cigarettes [51,52,53]. The sugars in electronic cigarette aerosols are the primary source of acrolein for users of these devices. Vreeke et al. [52] pay attention to the aerosol composition and the acrolein content. Triacetin, commonly used in producing e-liquids, has been shown to significantly increase the levels of acrolein, formaldehyde hemiacetals, and acetaldehyde in electronic cigarette aerosols [52].

Highly valuable data on the content of sugars and their impact on the concentration of acrolein in e-cigarettes is provided by Fagan et al. [49]. The researchers quantified the levels of sugars and aldehydes in e-cigarette liquids of 66 different brands differing in taste and concentration of nicotine and, at the same time, showed high concentrations of these compounds. As the authors emphasize, what is extremely important is that none of the ingredients tested in the work were listed on the product labels, which may lead to erroneous perceptions by consumers of the safety, harmfulness and ingredients of the liquids for unheated or heated e-cigarettes [49].

Therefore, the fundamental question remains—are e-cigarettes healthier than traditional cigarettes in the context of smokers’ exposure to harmful substances, especially acrolein? Studies in people who use e-cigarettes clearly show reduced levels in their urine of the primary metabolite of acrolein—3-hydroxypropylmercapturic acid (3HPMA) compared to users of combustible cigarettes, which indicates a reduced exposure of the body to acrolein. At the same time, however, it is worth noting that the observed level of the 3HPMA metabolite in the urine of electronic cigarette users was still very high, significantly higher than in the case of non-smokers. Although e-cigarettes are likely to be less carcinogenic than tobacco cigarettes, the study results indicate that e-cigarette users have an increased risk of exposure to acrolein compared to non-smokers and are still exposed to its harmful effects [41,43,54]. In addition, Cheng et al. [44], in studies using buccal cells isolated from e-cigarette smokers, clearly indicate a significant increase in the level of a carcinogen acrolein-DNA adduct 8R/S-3-(2′-deoxyribos-1′-yl)-5,6,7,8-tetrahydro-8-hydroxypyrimido [1,2-a]purine-10-(3H)-one (γ-OH-Acr-dGuo) compared to non-smokers. The authors thus emphasize the strong need for further research on the potentially toxic and carcinogenic effects of e-cigarette use, which result, among others factors, from the action of acrolein [44]. In recent studies, Lorkiewicz et al. [41] draw attention to another marker in the study of acrolein levels in e-smokers, 2,3-dihydroxypropylmercapturic acid (23HPMA), which, as they emphasize, is characterized by a much higher specificity of action. The level of 23HPMA increases in e-cigarette users while it remains unchanged in tobacco smokers [41].

Admittedly, very few studies directly indicate the acrolein-dependent toxicity of e-cigarettes. Nevertheless, it is significant that e-smokers exhibit similar homeostasis disorders identical to those accompanying exposure to acrolein. A growing body of evidence shows that e-cigarettes are responsible, among others, for disturbances in the balance of metalloproteinases, mucin production, and induction of inflammation [55].

## 6. Acrolein Is One of the Main Risk Factors for Chronic Obstructive Pulmonary Disease

Chronic obstructive pulmonary disease (COPD) is an incurable, progressive respiratory system disease caused by smoking (80–90% of cases) or by exposure to irritants, e.g., dust and chemicals. This disease is a huge global health problem. The World Health Organization (WHO) estimates that by 2030 COPD will be the third leading cause of death worldwide [56]. COPD is characterized by a poorly reversible, progressive decrease in airflow through the airways, accompanied by bronchitis, sputum production, emphysema, wheezing, and exertional dyspnea [57]. Unfortunately, despite the continuous development of medicine for COPD, there is no cure that is fully effective in inhibiting the progression of the disease and restoring full lung function. Pharmacological treatment of chronic obstructive pulmonary disease involves taking bronchodilators and anti-inflammatory drugs that relieve symptoms [58,59,60].

COPD most often affects cigarette smokers, as well as people from their immediate environment who regularly inhale tobacco smoke passively. One of the most significant risk factors for COPD is believed to be acrolein. Due to its broad spectrum of harmful biological effects, including interaction with proteins, regulation of gene expression of pro-inflammatory factors, pro-oxidative properties, and immunomodulatory properties, acrolein is at the molecular basis of this disease [15,61]. The most important molecular targets of acrolein related to its participation in the pathogenesis and development of chronic lung diseases and cancers of the respiratory system are summarized in Table 1.

Among all α,β-unsaturated aldehydes, acrolein is the strongest electrophile and therefore has an extreme reactivity with nucleophiles. Through the C-3 carbon, it interacts with the nucleophilic side of numerous proteins and DNA, creating adducts, i.e., the sulfhydryl group of cysteine, the imidazole group of histidine, the e-amino group of lysine and the guanidine group of arginine, or as in the case of DNA, the deoxyguanosine moiety. Preferably, acrolein forms Michael-type adducts on cysteine residues. This reaction is referred to as “protein carbonylation” as a result of the attachment of the carbonyl group of the aldehyde to the structure of the peptides [62,63,64]. The structural change of proteins often entails dramatic consequences in the form of a change in their function or its complete loss [65]. In addition, acrolein adducts formed on the thiol side of the cysteine chain are more stable than adducts formed by other unsaturated aldehydes. Due to the covalent interaction of acrolein with proteins, there is a low probability of systemic distribution of adducts. Therefore, their side effects are characterized by cytotoxicity directly at the site of formation [12]. As indicated by Avezov et al. [66], acrolein contained in cigarette smoke caused the carbonylation of several proteins in HaCaT cells in a dose- and time-dependent manner [66].

Over the past few decades, based on the analysis of the available literature, it can be undeniably stated that the basis of the harmful effects of acrolein in the pathology of many diseases, i.e., neurodegenerative diseases, heart diseases, cancer or lung diseases, is the aldehyde’s ability to extremely strongly interact with numerous proteins of our body that perform various regulatory, homeostatic or immunological functions [13,15]. The latest research by Chen et al. [67] indicates that acrolein interacts with α-Lys-16, α-His-50, and β-Lys-59 of smokers’ hemoglobin by changing the protein structure. The increase in the levels of acrolein-modified peptides in the hemoglobin of smokers undoubtedly indicates the multidirectional harmful effects of the aldehyde, which still requires better understanding [67]. Undoubtedly, the range of acrolein-dependent implications for the functioning of our cells, genes and immune system is unprecedented. In Figure 1, we have shown the most important effects of cytotoxic, immunomodulatory and gene-associated expression effects of acrolein.

### 6.1. Acrolein Cytoxicity

An extremely interesting feature of acrolein is the ease and diversity of its molecular mechanisms of biological activity in the pathogenesis of many diseases. Although acrolein has a chemically uncomplicated structure, the range of its biological activity is enormous [68].

Numerous studies have shown that acrolein is highly cytotoxic to many cell types in animal models and human lines [69,70]. The target organs of acrolein toxicity are primarily local tissues, the cells of which are directly exposed to acrolein [71]. Inhalation of acrolein causes irritation and inflammation of the airways, followed by hyperplasia and metaplasia of the respiratory epithelium [36,37]. In the case of smokers, the cells of the respiratory system, nasal epithelial cells [72], lung epithelial cells [73], bronchial epithelial cells [74,75] or human umbilical vein endothelial cells [69] are particularly exposed to the toxic effects of acrolein.

The molecular basis of shaping the cytotoxicity of acrolein undoubtedly involves free radicals, often generated by aldehyde, and the ability of acrolein to modulate gene expression or immunomodulatory properties [76]. The latest research also provides crucial information by Tulen et al. [77], which indicates that the key to the cytotoxicity of acrolein is its interaction with mitochondria. The authors show that acrolein dose-dependently disrupts the molecular regulation of mitochondrial metabolism in rat lungs, including transcript abundance of both subunits of the electron transport chain complexes and regulators associated with mitochondrial biogenesis [77]. These results are consistent with earlier studies [78], where it was shown that acrolein caused dose-dependent inhibition of components of the respiratory chain protein machinery, including complexes I and II, pyruvate dehydrogenase and α-ketoglutarate dehydrogenase. Extremely valuable information on the effect of aldehydes in cigarette smoke on mitochondria lung epithelial cells is provided by the latest review by Tulen et al. [79].

Moreover, as Conklin et al. [80] emphasized, the sensory irritant receptor, Transient Receptor Potential Ankyrin 1 (TRPA1), plays a crucial role in acrolein-dependent cytotoxicity. Animals deficient in TRPA1 or treated with a TRPA1 inhibitor were resistant to acrolein-induced pulmonary injury and survived severely toxic acrolein exposures at higher rates than wild-type mice [80].

Due to its biological multiactivity, acrolein-mediated cell death has been shown to occur via multiple pathways. In the case of high concentrations of acrolein, necrotic death of respiratory tract cells was observed. The activation of necrotic death leads to a feedback loop where necrosis-induced inflammation leads to more necrosis and oxidative damage to neighboring cells and vice versa [81,82]. Several studies using cigarette smoke indicate that it inhibits apoptosis and causes necrosis of human cells’ umbilical vein endothelium [83]. Acrolein-dependent necrotic cell death was confirmed using basal bronchial epithelial cells, indicating CS caused necrosis instead of apoptosis [84]. Comer et al. [72] indicate that acrolein in high doses (50 uM) and 4-h exposure caused mostly necrosis in human nasal epithelial cells [72].

As emphasized by several groups, acrolein also triggers the apoptosis pathway, primarily in the mitochondrial pathway [69,73,75]. In cells exposed to acrolein, key events characteristic of the programmed cell death process were observed—disturbances in the membrane potential of mitochondria, cytochrome c outflow, shifting the balance towards pro-apoptotic proteins of the Bcl-2 family, especially the Bax/Bcl-2 ratio, increased activity of caspases, including the effector caspase 3. In the latest studies, Liu et al. [56,62] indicate that the activation of acrolein-dependent apoptosis undoubtedly results from the generation of ROS, accompanied by increased levels of DNA damage-related indicators 8-hydroxy-2 deoxyguanosine (8−OHdG) and double-strand breaks, blocking the cell cycle and p38 MAPK and activation of c-Jun N-terminal kinase signaling pathways in response to oxidative stress [69,75].

### 6.2. Acrolein and Dysregulation of Mitochondrial Homeostasis

A relatively new point of view on the involvement of acrolein in lung dysfunction in the pathogenesis of COPD is its effect on the dysregulation of lung mitochondrial homeostasis. The negative effect of acrolein on the functioning of lung cell mitochondria occurs at several levels of the biological activity of these organelles. Acrolein-dependent mitochondrial abnormalities may concern their morphology, metabolism of energy processes, mitochondrial DNA damage or regulation of mitochondrial biogenesis and mitophagy. Studies show that acrolein exposure can disrupt the regulation of mitochondrial function and mitochondrial metabolism, inhibit mitochondrial bioenergetics, or even influence the fate of mitochondria through autophagy/mitophagy in airway and lung cells, and thus, lead to lung disease [77,85,86,87,88].

Beneficial results on the effect of acrolein on mitochondria were provided by Wang et al. [85]. Using human lung fibroblasts and alveolar epithelial cells, the authors showed that exposure to acrolein causes several changes in the functioning of mitochondria. Disturbances were already visible in the abnormal morphology of these organelles (e.g., fragmentation). In addition, it turns out that acrolein can generate damage not only to genomic DNA but also to mitochondrial DNA (mtDNA) and can alter mtDNA copy numbers in human lung epithelial cells and fibroblasts [85].

It turns out that acrolein can also disrupt the regulation of mitochondrial biogenesis and mitophagy [77,85,88]. After exposure to acrolein, a decrease in the expression of PPARGC1A and its downstream transcription factors involved in mitochondrial biogenesis was observed in vivo in rats [77] and in vitro in lung fibroblasts [88]. Moreover, as indicated by Wang et al. [85], exposure to acrolein induced mitochondrial fission in human lung fibroblasts, followed by autophagy/mitophagy. As the authors suggest, acrolein-dependent mtDNA damage may be crucial here, resulting in the loss of mtDNA through mitochondrial fission and mitophagy [85]. In addition, the impact of acrolein on mitochondrial dynamics was also carried out by modulating the transcriptional regulation of mitochondrial fission and fusion [77,85].

As demonstrated by in vivo studies in rats, exposure of animals to acrolein caused an aldehyde-dependent change in mitochondrial functionality and energy metabolism in the airways and lungs [86]. Acrolein contributed to switching the metabolism towards glycolysis, as evidenced by the increased level of lactates and the increase in the activity of glycolytic enzymes. In addition, a decrease in the amount of mRNA-encoding proteins constituting complexes of the electron transport chain was simultaneously observed [77,86]. These findings are consistent with studies by Luo et al. [88] on human lung fibroblasts. Exposure to acrolein caused in cells, apart from disturbances of the mitochondrial potential, a significant decrease in the expression of complexes I, II, III and IV of the mitochondrial electron transport chain [88].

Acrolein-dependent change in mitochondrial metabolism was also confirmed in vitro after exposure to alveolar epithelial cells. As emphasized by Agarwal et al. [87], acrolein caused a disturbance of the energy homeostasis of the mitochondria and the transition from the glycolytic pathway to the pentose phosphate pathway, which had its consequences in the form of using palmitate from phosphatidylcholine as an alternative substrate for mitochondrial respiration. Importantly, as the authors indicate, this may cause a decrease in the level of surfactants in the lungs, which is characteristic of the pathogenesis of COPD [87].

### 6.3. Acrolein-Dependent Oxidative Stress in Smokers

In the case of smokers of tobacco products, cigarette smoke and the harmful substances contained therein (including acrolein) are the primary source of free radicals. Reactive oxygen species and lipid peroxidation play an important role in the pathogenesis of cardiovascular and lung diseases such as coronary artery disease and chronic obstructive pulmonary disease [12].

The theoretical basis for the toxic effects of acrolein and its potential mechanism of oxidative stress induction is provided by a recent study on Caenorhabditis elegans by Hong et al. [89]. Excessive acrolein-dependent production of ROS is associated with a decrease in the activity of ROS scavenging enzyme superoxide dismutase (SOD) and catalase (CAT) and concomitant activation of oxidative-stress-related pathways DAF-16/FOXO. The research undoubtedly contributed to enriching the toxicological evaluation of acrolein [89].

With regard to lung diseases, including COPD, the influence of acrolein on oxidative stress and the generation of free radicals is carried out at several molecular levels, starting with the direct interaction of the aldehyde with the enzymes involved in oxidative homeostasis, through to the influence on mitochondria, to the regulation of signaling pathways and genes of which protein products are involved in oxidative regulation. As emphasized by Yasuo et al. [90], significantly elevated levels of acrolein in plasma collected from COPD patients and lung tissues were closely involved in the pathogenesis of the disease through interference in the balance of oxidative stress versus antioxidant potentiality [90]. Abundant evidence indicates that one of the main targets of acrolein is proteins with antioxidant properties. One such molecule is glutathione (GSH) which plays a crucial role in cellular reductive processes and detoxifying harmful ROS. As a result of the interaction of acrolein with GSH, GSH-acrolein adducts are formed, the cellular GSH pool is inactivated, and the tripeptide is unable to neutralize ROS. These findings might be a biochemical mechanism of acrolein-induced toxicity that has been found in patients with COPD [91,92,93]. In vitro studies on a human airway tissue model showed an acrolein dose- and length-dependent depletion of GSH and a consequently disrupted redox balance [14].

Moreover, as the authors underline, changes in heme oxygenase-1 (HMOX-1) protein expression in response to exposure to acrolein are correlated with changes in the GSH/GSSG ratio. In conclusion, this indicates the effect of acrolein on redox homeostasis disorders and the induction of oxidative stress [14]. The innovative in vitro dosimetry analyses for acrolein exposure in normal human lung epithelial cells and human lung cells were conducted by Xiong et al. [94]. Acrolein in normal human bronchial epithelial (NHBE) cells and human mucoepidermoid pulmonary carcinoma (NCI-H292) cells has been shown to significantly induce protein carbonylation, GSH depletion and formation of GSH-acrolein adducts (GSH-ACR). Normal NHBE cells were significantly more sensitive to acrolein exposure and its harmful effects. Moreover, as the authors point out, GSH depletion and conjugation are the main adverse events directly related to acrolein-induced cytotoxicity [94].

GSH is not the only molecule involved in redox signaling, which is a direct target for acrolein. The aldehyde interacts with amino acid residues of other key proteins for protection against ROS, including thioredoxins (Trxs), peroxiredoxins (Prxs), and thioredoxin reductase, leading to their dysfunction [95,96]. As in the case of GSH, Spiess et al. [97] emphasize that acrolein causes enzyme inactivation in a dose-dependent manner by forming adducts with Trx.

It is known that oxidative stress generated by acrolein in cigarette smoke affects lung cells and endothelial vascular cells. Highly valuable studies by Horinouchi et al. [98] indicate that acrolein in human vascular endothelial cells induces oxidative stress by affecting NO production, by reducing the activity and total level of the eNOS protein. These findings suggest that cigarette smoking leads to the attenuation of NO-dependent endothelium-derived vasodilation [98].

Oxidative damage to lung cells in COPD is also caused by the upregulation of genes directly involved in redox homeostasis. Several articles indicate that acrolein modulates the expression of genes whose protein products are involved in the production of ROS. Sun et al. [99] emphasize that in vitro and in vivo, they observed the enhancement of the acrolein-induced production of ROS and lung injury by GADD34 (Ppp1r15a) with simultaneous inflammation. GADD34 was highly expressed under the stimulation of acrolein, indicating that GADD34 might be involved in the pathogenesis of alveolar injury by producing ROS. As the authors indicate, GADD34 may be one of the key proteins in acrolein-induced lung inflammation and tissue injury [99]. Moreover, acrolein can alter upstream and downstream components of a signaling cascade, including transcription factors that modulate the expression of genes related to redox homeostasis. The nuclear erythroid-2 related factors 2 (Nfr2) signaling pathway is also affected by acrolein. The aldehyde forms adducts with this member of the basic leucine zipper transcription factor family, the nuclear erythroid-2 related factors 2 [100,101,102]. The Nrf2 pathway plays an important role in the regulation of genes that control the expression of proteins critical to the detoxication and elimination of ROS. The nrf2 signaling pathway is involved, among others, in the protection of cells after exposure to cigarette smoke in the respiratory system and ocular epithelium [103].

It is also key that acrolein affects another transcription factor, the nuclear factor NF-κB. This protein is involved in regulating the expression of over 400 genes in our body that control processes related to the cell cycle, cell death, immune response and many others [104,105]. The mechanisms of using NF-κB and the complex signaling pathways by acrolein in the course of COPD are very complex and not fully understood. What is certain is that depending on the degree of cell exposure to acrolein, the genetic response varies. In the case of acute exposure and high doses of acrolein, inhibition of the expression of pro-inflammatory cytokines is observed, which affects susceptibility to infections. On the other hand, chronic exposure to acrolein in low doses results in a significant increase in pro-inflammatory factors and cell and tissue damage, including that caused by ROS. The key to oxidative stress resulting from the relationship in the interaction of acrolein NF-κB are proteins directly involved in the immune response. Thus, acrolein modulates gene expression and the production of specific proteins regulating free radical reactions in such a way as to initiate oxidative stress and inflammation in the lower respiratory tract as much as possible [13,73,76,106].

Of course, the free radicals generated depending on acrolein are not without consequences for the proper functioning of lung cells and the respiratory system of smokers. Disturbances of redox homeostasis lead to a number of further morphological and structural changes in cells. The latest cohort studies indicate that the formation of ROS in smokers is followed by lipid peroxidation and, consequently, a violation of the integrity of cell membranes, oxidative DNA damage and the resulting cancer mutations, or chronic inflammation within the lung cells and damage to the epithelium [107,108].

### 6.4. Mucus Hypersecretion

Normally, mucins are gradually released into the respiratory tract, where they absorb water and form a thin layer of protective mucus that fights pathogens and is easily removed by the cilia. In COPD, large amounts of mucins are rapidly released as a result of inflammation, resulting in thick mucus that can impair lung function [109,110].

Cigarette smoke can directly increase the expression of mucin genes, mainly MUC5AC and MUC7, and to a lesser extent MUC1 and MUC2 [111,112], and additionally, can synergistically increase the response to pro-inflammatory cytokines and bacterial infections [113]. Among the many components of tobacco smoke, acrolein is considered one of the most potent stimulants of excessive mucus secretion, the main feature of COPD [114]. In animal models exposed to acrolein, there was damage to the airway epithelium, bronchiolitis, mucosal metaplasia, and increased MUC5AC expression accompanied by excessive accumulation of macrophages in the lungs [115,116]. As numerous studies indicate, the main direct target for acrolein is primarily the MUC5AC gene, the expression of which increases after exposure to low concentrations of acrolein [14,115,117,118].

Importantly, evidence shows that e-cigarettes also dysregulate mucin expression. E-cigarette users show increased mucin expression in human bronchial epithelial cells [119] and induced sputum [120]. Decreased mucus secretion velocity [121] and increased mucin expression (MUC5AC) have been reflected in animal models of e-cigarette exposure [119,122].

It should be emphasized, however, that acrolein affects the production of mucins not only by directly affecting the expression of the relevant genes but also indirectly through mechanisms involving the activation of metalloproteinases and the activation of EGFR-mediated activation of ERK1/2, JNK and p38 MAPK signaling pathways, as shown by studies on transgenic rodent models and human cell lines in vitro [117,123,124,125,126,127]. Typically, the main metalloproteinases activated by cigarette smoke are metalloproteinase-1, -2, -8, -9 and -14 [13]. In a recent study, Xiong et al. [14] showed that acrolein additionally modulated the activity of metalloproteinases-7, -10, -12 and -13. Acrolein, by activating metalloproteinases (including -9 and -14), indirectly stimulates the activation of EGFR-MAPK signaling, which in turn affects the production of mucins dependent on this pathway. Chen et al. [128] demonstrated in a rat research model that EGFR/ERKA1/2-mediated mucin hyperproduction after exposure to acrolein may be a consequence of Ras GTPase activation [128].

Collectively, the evidence shows that exposure to cigarette smoke from both traditional tobacco products and e-cigarettes interferes with the normal mucociliary protection against pathogen invasion that is often seen in COPD. The stimulating effect of acrolein on mucin production (directly or indirectly) may promote chronic mucus secretion in COPD.

### 6.5. Acrolein-Dependent Tissue Modification and Degradation in Emphysema

In addition to obstructive bronchiolitis, the most important pathology found in COPD is emphysema, the irreversible destruction of the gas exchange surface of the lungs, caused by the destruction of the interstitial tissue of the lungs and the loss of interalveolar septal attachments to the small airways. The process affects the entire lungs; their elasticity deteriorates, they contain an increased amount of air, and the course of gas exchange is impaired. The leading cause of emphysema in COPD is an imbalance between the proteolytic enzymes that break down connective tissue components and anti-proteinases [129].

During acute lung injury, pulmonary oedema occurs, caused by increased mucosa and endothelium epithelial permeability and decreased clearance of oedema fluid through the alveolar epithelium. Both epithelial and endothelial barriers are maintained, among others, by tight junction proteins—claudins. One of the targets for acrolein is the CLDN5 gene encoding claudin 5. It has been shown that acrolein-dependent regulation of CLDN5 expression in mice impairs the epithelial barrier function and induces epithelial cell death that leads to lung injury [130]. Importantly, as emphasized by Chen et al. [131], the weakening of intercellular connections by acrolein also occurs in intestinal endothelial cells, leading to intestinal barrier dysfunction and permeability and, consequently, endoplasmic reticulum stress-mediated cell death [131].

Another critical effect of acrolein on lung tissue is the remodeling of the extracellular matrix (ECM). ECM remodeling significantly impacts organ function, including lung function in COPD. Acrolein-induced tissue remodeling is caused by insufficient repair after cell loss and damage to the extracellular matrix mediated by inflammation. Acrolein affects two essential aspects of ECM cell interaction critical for respiratory tissue repair: cell attachment to the ECM and ECM remodeling. The loss of elasticity results from the replacement of elastin, an essential component of the lung’s ECM, with collagen, a much less elastic fibre [132].

Acrolein-dependent dysregulation of metalloproteinase activity is not only associated with mucus overproduction in COPD but also has other serious consequences for the pathogenesis of this disease. It has been proven that in addition to cytokines, metalloproteinases also play a crucial role in the remodeling of the respiratory tract in smokers. Several MMPs, mainly MMP-1, -2, -8, -9, -12 and -14, can be activated by cigarette smoke and have been implicated in the destruction of lung tissue and thus associated with the development and exacerbation of COPD as well as emphysema [14,133,134]. As indicated by in vitro and in vivo studies, exposure to acrolein may lead to unbalanced activation of MMPs by activating the EGRF-, MAPKs- and mTOR pathways or a to decrease in the activity of some MMP inhibitors [124,126,127,135].

In airway smooth muscle (ASM) cells isolated from human lungs, it was shown that cigarette smoke (CS) caused a dose-dependent increase in the deposition of the alpha 1 chain of type VIII collagen (COL8A1) as well as increased expression of matrix metalloproteinase MMP-1, MMP- 3 and MMP-10. In addition, the tendency for increased adhesion of ASM cells in COPD patients was reduced by CS, as was the ability of ASM cells to heal wounds when exposed to the highest concentration of CS [136]. Deshmukh et al. [117,124], in their studies, indicate that at concentrations that can be found in the sputum of COPD patients, acrolein can increase the expression of matrix metalloproteinases 9 and 14. Exposure to acrolein in vivo increased the activity of MMP-9 and MMP-14 and mucin transcript and protein levels. At the same time, a decrease in MMP-9 inhibitor transcript activity was observed in mouse lung tissue [117,124]. Chaudhuri et al. [137] noted that MMP-12 activity was higher in smokers with COPD. The authors suggest that elevated levels of MMP-12 directly correlate with the severity of the disease. Importantly, research supports the notion that unbalanced proteolytic/anti-proteolytic activity may contribute to the pathology of COPD [137]. In turn, Hunninghake et al. [138] indicate that MMP-12 may be associated with the degradation of structural proteins in the lungs, including elastin, which results in a reduction in forced expiratory volumes and thus is a risk factor for COPD [138].

Importantly, recent research also indicates that exposure to e-cigarettes also affects the proteinase–antiproteinase balance in the lungs. A human cohort study demonstrated that MMP-2, -8 and -9 levels were significantly elevated and even comparable to cigarette smokers [120,139]. Studies on the murine model also showed dysregulations in proteinases, where after four months of exposure to e-cigarette vapor, an increase in the level of MMP-9 and -12 was observed [122].

### 6.6. Immunomodulatory Properties of Acrolein

Underlying the pathogenesis of COPD are inflammatory and immune responses to inhaled toxic compounds from tobacco smoke. Exposure to tobacco smoke affects the immune system, impairing the ability to proper immune and anti-inflammatory responses. Consequently, smoking and COPD are associated with bacterial colonization of the airways, resulting in greater airway inflammation and an accelerated decline in lung function [140]. The latest research by Endo et al. is extremely interesting as it sheds entirely new light on the action of acrolein in the context of its influence on our immune system. Studies have confirmed the presence of innate B cells expressing the acrolein-specific IgM-B cell receptor, which suggests that acrolein, in addition to its traditionally considered toxic aldehyde function, may play a role as a trigger of the innate immune response [141].

Acrolein, as a component of tobacco smoke, has a solid immunomodulatory effect, as demonstrated in numerous studies [13,14,72,73,142,143]. In particular, acrolein is able to interfere with biological macrophagic functions, including phagocytosis, adhesion or homodimerization of Toll-like receptors [144,145]. The complexity with which acrolein affects the modulation of the immune response is remarkable because it can act as both a pro-inflammatory and anti-inflammatory agent. Reports of acrolein-dependent changes in inflammatory signaling and expression of genes involved in the immune response are varied. They suggest a significant, multi-level complexity of the immunomodulatory properties of acrolein. As indicated by Burcham et al. [146], the critical factors for the immunogenic properties of acrolein are its dose, degree of host cell exposure and, undoubtedly, its type [146]. The pro- or anti-inflammatory effect of acrolein depends on the aldehyde control of the expression and activity of important pro-inflammatory factors and gene expression regulators, especially interleukins, including IL-8 and NF-κB [14,72,73,76,142,143].

The inhibition of the immune response by reducing the defense against viral and bacterial infections with the participation of acrolein takes place through several pathways. The altered immune response resulting in a significant reduction of inflammation occurs primarily due to the suppression of the acrolein-mediated factor NF-κB [106,147,148]]. Acrolein, by direct alkylation of Cys-61 cysteine and Arg-307 arginine residues in the NF-κB (p50) binding domain, causes a significant inactivation of the transcription factor and weakens its binding to the promoter sequences of IL-2, IL-10, TNF-α, and INF-γ and, consequently, a decrease in those pro-inflammatory mediators/cytokines that are predominantly regulated in the NF-κB pathway [145,148,149].

In contrast to immunosuppressive effects, acrolein is demonstrated to induce pro-inflammatory mediators via multiple pathways in both in vivo and in vitro studies [14,72,73,76,142,150]. The pro-inflammatory properties of acrolein are greatly influenced by the aldehyde’s ability to activate the expression of NF-κB and to activate the signaling pathways dependent on this transcription factor and, consequently, the release of inflammatory mediators such as IL-1β, IL-6, IL-8, TNF-α, and IFN-γ [14]. In rat lung endothelial cells, a significant increase in cyclooxygenase-2 (COX-2) involved in the later stages of the inflammatory reaction in the NF-κB activation pathway was observed after exposure to acrolein [151]. The research by Sun et al. [73] confirms these reports. As the authors indicate, acrolein induced the inflammation of murine epithelial cells in the NF-κB activation pathway. In addition, there was a marked increase in CD11c+F4/80high macrophages in the cells and in the release of pro-inflammatory cytokines/factors by these cells. It should also be mentioned that ROS generated by the aldehyde played a significant role in the pro-inflammatory properties of acrolein. Disruption of redox homeostasis is one of the causes of chronic inflammation, which ultimately leads to cell damage and death [73]. Additionally, as outlined by several research groups, metalloproteinases and ROS activated by acrolein are also involved in the pro-inflammatory properties of acrolein [127,143]. In addition to activating NF-κB-dependent pathways, acrolein is an activator of complex kinase pathways, such as p38 MAPK/MK2 [76] or ERK 1/2 [150], which, for example, modulate the body’s response to pathogens by triggering the expression of pro-inflammatory factors, including IL-8. Moretto et al. [76,150], in studies on human pulmonary cells and lung fibroblasts, showed that acrolein-dependent activation of these signal kinases increased IL-8/CXCL8 mRNA stability and thus increased IL-8 protein expression and consequent severe inflammation.

As indicated by recent studies by Takamiya et al. [152,153], the immunomodulatory properties of acrolein are manifested not only by influencing the signaling pathway and regulating the expression of genes of inflammatory mediators, but also by direct impact on proteins involved in the immune response. Surfactant proteins A and D (SP-A/SP-D) belong to the family of collectin proteins (collagen-containing C-type lectins), which play an essential role in the functioning of the innate immune system. SP-A and SP-D are involved in the immune response, especially in the lungs [154]. Experiments on lung tissues isolated from mice exposed to cigarette smoke showed that acrolein interacted directly with SP-A and SP-D, causing changes in their amino acid residues. These modifications led to disturbances in the multimer of the SP structure, which in turn significantly contributed to the reduction of their capacity in the proper immunological response in the lungs—weakening the inhibition of bacterial growth and promoting the activation of macrophages and the phagocytosis of pathogens [152,153].

**Table 1 cells-12-00879-t001:** Cellular and molecular targets of acrolein related to its molecular mechanisms of activity in chronic lung diseases and respiratory cancer.

Target	Experimental Approach	Effect	References
**Cytotoxic Properties of Acrolein**			
**In Vitro Studies**			
Human umbilical vein endothelial cells (HUVEC)	Acrolein exposure	↑ ROS, ↑ 8-OHdG—DNA damage, ↑ Bax, ↓ Bcl-2, ↑ caspase-3—apoptosis	[69]
Primary nasal epithelial cell cultures (PNECs) from healthy non-smokers	Cigarette smoke extract	↑ caspase-3—apoptosis,	[72]
Bone marrow-derived GM-CSF-dependent immature macrophages (GM-IMs)	Acrolein exposure	↑ ROS, ↑ caspase-3—apoptosis	[73]
Lewis lung carcinoma (LLC) cells			[73]
Human bronchial epithelial cell line (HBE1)	Acrolein exposure	↓ GSH, ↑ ROS, phosphatidylserine (PS) externalization and DNA fragmentation—apoptosis	[74]
Human bronchial epithelial cell line (BEAS-2B)	Acrolein exposure	↑ ROS, ↑ 8-OHdG, ↑ γ-H2AX—DNA damage, ↑ Bax, ↓ Bcl-2, ↑ caspase-3—apoptosis, G2/M cell cycle blockade	[75]
Murine FL5.12 pro B lymphoid progenitor cells	Acrolein exposure	↓ Caspase-9, -8 and -3—apoptosis inhibition, oncotic/necrotic response	[81]
Human lung adenocarcinoma cells (A549)	Acrolein exposure	↑ ROS, cytochrome c release, ↑ Bax, ↑ Bad, ↓ Bcl-2, ↑ apoptosis inducing factor AIF, ↑ caspase-9, -7, -6, -3—apoptosis	[82]
Human lung adenocarcinoma cells (A549), Human umbilical vein endothelial cells (HUVEC)	Cigarette smoke condensate	↓ Caspase-9, and -3 —apoptosis inhibition, necrotic response	[83]
Human tracheobronchial epithelial cells (hTBE)	Cigarette smoke extract	↓ Caspase-7, and -3 —apoptosis inhibition, necrotic response	[84]
Human alveolar epithelial type I-like (ATI-like) cells	Cigarette smoke extract	↑ Caspase-7, and -3- apoptosis, necrotic response	[103]
**In vivo studies**			
Lungs isolated from male albino rats	Acrolein inhalation	↑ Proinflammatory TNF-α, ↑ Il-6 ↑ Bax, ↓ Bcl-2—apoptosis	[70]
**Dysregulation of mitochondrial homeostasis**			
**In vitro studies**			
Normal human lung fibroblasts (MRC-5), lung adenocarcinoma cells (A549)	Acrolein exposure	↓ PARP-1, ↑ ROS, ↑ Caspase-9, and -3 —apoptosis, ↓ mitochondrial membrane potential, ↓ ATP, ↑ mtDNA damage, ↑ mitochondrial fission and mitophagy	[85]
Lung alveolar cell line RLE-6TN, Clara cell-like human bronchialepithelial cell line (H441), pAT2 cells isolated from male A/J mice (Jackson Laboratories)	Acrolein exposure	↑ Mitochondrial metabolism of palmitate, ↑ glucose-6-phosphate dehydrogenase activity, ↑ phospholipase A2, ↓ phosphatidylcholine, inhibition of glycolysis	[87]
Human lung fibroblast cells (IMR-90)	Acrolein exposure	↑ β-galactosidase, ↑ p53, ↑ p21, ↓ DNA synthesis, ↓ Sirt1, ↑ ROS, ↓ mitochondrial membrane potential, ↓ mitochondrial biogenesis regulator PGC-1, of subunits of the electron transport chain complexes	[88]
**In vivo studies**			
Lung tissue isolated from male Wistar rats	Acrolein inhalation	↓ Transcript abundance of subunits of the electron transport chain complexes, ↓ PPARGC1A regulators associated with mitochondrial biogenesis	[77]
SM/J (sensitive) and 129 × 1/SvJ (resistant) inbred mouse strains	Acrolein inhalation	↓ Transcript abundance of subunits of the electron transport chain complexes, ↑ activity of glycolytic enzymes	[86]
**Redox homeostasis imbalance, carbonylation of antioxidant enzymes**			
**In vitro studies**			
Human gingival fibroblasts cells (HGFs)	Cigarette smoke	↑ ROS, ↑ protein carbonylation, ↑ GSH-acrolein adducts, ↓ -SH groups	[91]
Lung adenocarcinoma cells (A549)	Cigarette smoke extract	↑ protein carbonylation, ↑ GSH-acrolein adducts, ↓ -SH groups	[92,93]
Human ALI airway tissue models	Acrolein exposure	↑ protein carbonylation, ↑ GSH-acrolein adducts, ↑ HMOX-1	[14]
Normal human primary bronchial epithelial cells (NHBE), human mucoepidermoid pulmonary carcinoma cells (NCI-H292)	Acrolein exposure	↑ protein carbonylation, ↑ GSH-acrolein adducts	[94]
Human bronchial epithelial cell line (BEAS-2B)	Acrolein exposure	↑ Trx1 and Trx2 oxidation, ↑ peroxidases Prx1 and Prx3 oxidation, ↓ TrxR,	[95]
Human bronchial epithelial cell line (HBE1)	Acrolein exposure	↑ protein carbonylation, ↑ GSH-acrolein adducts, ↑ Trx1 oxidation, ↑ Prx1 oxidation, ↑ GSH-acrolein adducts	[97]
Human umbilical vein endothelial EA.hy926 cells	Cigarette smoke extract	↓ eNOS, ↓ NO	[98]
Murine bone marrow-derived macrophage cells, Lewis lung carcinoma cells (LLC)	Acrolein exposure	↑ ROS, ↑ GADD34	[99]
Bovine aortic endothelial cells (BAECs)	Acrolein exposure	↓ GSH, ↑ ROS, ↑ Heme oxygenase-1 (HO-1), ↑ Nrf2 signaling pathway	[100]
Lung adenocarcinoma cells (A549)	Acrolein exposure	↓ GSH, ↑ Nrf2 signaling pathway	[101]
Human alveolar epithelial type I-like (ATI-like) cells	Cigarette smoke extract	↑ HO-1, ↑ Nrf2 signaling pathway, ↑ lipid peroxidation	[103]
**In vivo studies**			
Female wild-type C57BL/6 mice	Acrolein inhalation	↑ ROS, ↑ GADD34	[99]
**Mucus hypersecretion and tissue degradation in emphysema**			
**In vitro studies**			
Human pulmonary mucoepidermoid carcinoma cell line NCI-H292	Cigarette smoke extract	↑ MUC7, ↑ MUC5AC	[112,113]
Human pulmonary mucoepidermoid cells NCI-H292	Acrolein exposure	↑ MUC5AC, ↑ MMP-9, ↑ MMP-14	[123,124]
Human ALI airway tissue models	Acrolein exposure	↑ MUC5AC, ↑ MMP-7, -10, -12, and -13	[14]
Human aortic vascular smooth muscle cells	Cigarette smoke extract	↑ MMP-1	[126]
Primary human lung micro-vascular endothelial cells, fusion of human umbilical vascular endothelial cells with the lung carcinoma cell line A549	Acrolein exposure	↑ CLDN5	[130]
The murine macrophage cell line, J774A.1	Acrolein exposure	↑ MMP-9, ↑ 5-lipoxygenase	[135]
**In vivo studies**			
MUC7 transgenic mouse tissues	Cigarette smoke extract	↑ MUC7	[112]
Female Han Wistar rats	Cigarette smoke extract	↑ MUC5AC	[113]
Gene-targeted Mmp9(^-^/^-^) mice, Mmp9(^+^/^+^) mice,	Acrolein exposure	↑ MUC5AC, ↑ MMP-9	[117]
Specific pathogen-free grade male C57BL/6 Mice	Airway remodeling model—acrolein inhalation	↑ MUC5AC	[118]
Human bronchial epithelial cells and induced sputum from e-cigarette users	E-cigarette exposure	↑ MUC4, ↑ MUC5AC, ↑ MMP-8, ↑ MMP-9	[119,120]
A/J mice (Jackson Laboratories)	E-cigarette exposure	↑ MUC5AC	[122]
Pathogen-free male Kunming mice	Acrolein intraperitoneal injection	↑ MUC5AC, ↑ MMP-9	[125]
Inbred mouse strains	Acrolein exposure	↑ CLDN5, acute lung injury (ALI)	[130]
C57BL/6J mice	Cigarette smoke extract	↓ Elastin, ↑ collagen	[132]
Airway smooth muscle from smokers	Cigarette smoke extract	↑ MMP-1, -3 and -10, ↑ collagen VIII alpha 1 (COL8A1)	[136]
Sputum from smokers	Cigarette smoke extract	↑ MMP-12	[137]
Bronchoscopies on cigarette smokers, and e-cigarette users	Cigarette smoke extract, e-cigarette exposure	↑ MMP-2, ↑ MMP-9	[139]
**Immunomodulatory properties**			
**In vitro studies**			
Human ALI airway tissue models	Acrolein exposure	↑ IL-1b, ↑ IL-6, ↑ IL-8, ↑ TNF-α, ↑ IFN-γ	[14]
Primary nasal epithelial cell cultures (PNECs) from healthy non-smokers	Cigarette smoke extract (CSE) containing acrolein	↑ IL-8	[72]
Bone marrow-derived GM-CSF-dependent immature macrophages (GM-IMs), Lewis lung carcinoma (LLC) cells	Acrolein exposure	↑ NF-κB, ↑ TNF-α, ↑ IL-6, ↑ IL-12, ↑ CD11c+ F4/80 ^high^ macrophages	[73]
Human small airway epithelial cells (SAECs), normal human bronchial epithelial cells (NHBEs), normal human lung fibroblasts (NHLF)	Acrolein exposure	Influence on the p38MAPK/MK2 and ERK 1/2 signaling pathway, ↑ IL-8	[76,150]
Rat lung epithelial cells (LE)	Acrolein exposure	Influence on the Raf-1/ERK signaling pathway, ↑ NF-κB, ↑ COX-2	[151]
**In vivo studies**			
Lungs isolated from male albino rats	Acrolein inhalation	↑ TNF-α, ↑ Il-6	[70]
Male C57BL/6J mice, peripheral blood was drawn from healthy, non-smoking adult volunteers	Acrolein inhalation and exposure	↓ NF-κB, ↓ IL-2, ↓ IL-10, ↓ TNF-α, ↓ INF-γ	[145,148,149]
Female C57BL/6 mice	Cigarette smoke extract	↓ SP-A, ↓ SP-D	[152,153]

The effects of acrolein are marked as follows: ↓ decreased, ↑ increased.

## 7. Acrolein in the Development of Lung and Oral Cancer

According to IARC, acrolein is classified as a group 2A compound—probably carcinogenic to humans. Of course, there is plenty of evidence from in vitro, in vivo, and human cohort studies that indicate this aldehyde’s cancer potential [44,108,155,156]. Moreover, the latest research shows that acrolein, by affecting specific signaling pathways, is not only directly involved in mutagenesis but also contributes to increasing the resistance of cancer cells to traditional cisplatin chemotherapy [157]. Clear evidence for the carcinogenic and cytotoxic properties of acrolein is provided by the studies of Matsumoto et al. [158], where a 2-year inhalation of acrolein in both mice and rats induced, among others, squamous cell carcinomas in the nasal cavity.

It is practically impossible to draw a clear line between the molecular mechanisms of acrolein involved in COPD and lung cancer. As in the case of COPD, the critical activities of acrolein for mutagenesis are its ability to regulate the expression of specific genes, immunomodulatory properties, imbalance in ROS homeostasis and interaction with macromolecules, including DNA. As shown in Figure 2, the genotoxic properties of acrolein are realized on many levels of interaction with DNA.

Acrolein, a component of cigarette smoke, is considered one of the main risk factors for lung cancer [68]. Notably, as indicated by Peterson et al. [159] in the latest research, although acrolein itself is a potent mutagen, it can also synergize with other compounds present in cigarette smoke. Acrolein has been proven to increase the pulmonary tumorigenic activity of the nitrosamine 4-(methylnitrosamino)-1-(3-pyridyl)-1-butanone (NNK). Three-hour incubation of mice on acrolein did not induce lung adenomas, whereas it did significantly enhance NNK’s lung carcinogenicity; however, the mechanism of this interaction is unclear. Research undoubtedly provides valuable information that tobacco smoke chemicals are more active in a mixture, and that the complex interactions between them increase tobacco smoke’s strong carcinogenic properties [159].

The key to acrolein-dependent mutagenesis is its high reactivity allowing it to react with DNA. It has been shown that acrolein interacting with DNA can produce interchain cross-links of double-stranded DNA and DNA-protein cross-links [68]. The scope of action of acrolein on nitrogenous bases is very diverse. Acrolein can induce oxidative DNA damage due to ROS homeostasis disorders and free radical generation. Moreover, this aldehyde readily reacts with deoxyguanosine (dG), producing two exocyclic DNA adducts, α- and γ-hydroxy-1, N2-propano-2′-deoxyguanosine (a-HOPdG and c-HOPdG), but is also capable of forming adducts of 2′-deoxyadenosine and 2′-deoxycytidine DNA bases and thymidine [108,160,161]. The result of the formation of acrolein-DNA adducts is mutations within DNA fragments crucial to the fate of the cell. Like other polycyclic aromatic hydrocarbons, acrolein primarily causes G to T transversions and G to A transitions in the CpG-rich hotspots of the TP53 gene, a known mutation site in smoking-associated lung cancer [162]. It is worth noting that an increasing number of new studies indicate that the amount of acrolein-DNA adducts can be a very reliable indicator of the risk of developing cancer in smokers, but also, importantly, in e-cigarette users, both in the case of lung cancer [108,163] and oral cavity cancer [33,142]. The obtained results confirm a significantly higher amount of acrolein-DNA adducts in people using tobacco products compared to non-smokers [44,108,155,160,163].

It is also important to emphasize that the genotoxic properties of acrolein are not limited only to the interaction with DNA and damage to nitrogenous bases. This aldehyde causes significant inhibition of DNA repair systems, both nucleotide excision repair and base excision repair, by inactivating or weakening the enzymes crucial for repairing damaged DNA. Lack of repair or its slower activity allows for the successive accumulation of acrolein-DNA adducts and duplication of mutations [156,164,165].

Interestingly, the direct interaction of acrolein with DNA and the formation of adducts and the inhibition of the activity of DNA repair enzymes do not seem to be the only genotoxic mechanisms of acrolein. Notably, as outlined by Chen et al. [166] and Fang et al. [167], acrolein may act as an epigenetic agent by interacting with histones and consequently impairing chromatin assembly. The key to acrolein-dependent epigenetic alterations of gene expression seems to be its interaction with lysine residues, including lysine 5 and 12, on histone H4. The consequence of modifying histones with acrolein is the inhibition of their acetylation of N-terminal tails, modifications that are important for nuclear import and chromatin assembly [166,167].

## 8. Goals for the Future

A huge challenge for modern medicine is the development of effective methods and therapeutics that inhibit the harmful effects of acrolein. Due to the crucial role of acrolein in the pathogenesis and course of COPD, neutralizing the aldehyde may be one of the potential approaches for treating the disease. Acrolein-dependent oxidative stress plays a key role in the pathogenesis of COPD [168]. One of the main strategies to inhibit the harmful effects of acrolein is to inactivate its prooxidative activities. Great hopes are associated with using compounds containing thiol groups with antioxidant properties [169,170,171]. One of the most widely used acrolein scavengers is n-acetylcysteine (NAC). In the direct NAC–acrolein reaction, the aldehyde is inactivated to form covalent Michael adducts [172]. Several studies provide evidence of the beneficial effect of the NAC scavenger against cigarette smoke and acrolein in COPD [127,173,174].

Nevertheless, as indicated by Moitra’s recent research [175], the therapeutic effectiveness of NAC in treating COPD remains controversial, e.g., due to reduced oral bioavailability [175]. In addition, ineffective glutathione levels may be another important reason for not achieving the desired therapeutic effect [176]. Another strategy for fighting acrolein in COPD is to increase the activity of the Nrf-2 pathway involved in ROS protection and redox homeostasis. Natural polyphenols such as resveratrol [177,178] or curcumin [179] affect the activation of the Nrf-2 pathway. However, it should be remembered that the therapeutic effectiveness of many natural polyphenols is hampered by limited bioavailability in vivo and/or rapid inactivation in the gastrointestinal tract [180].

Although many compounds, both natural and drug, are effective at scavenging acrolein, some studies have shown that accumulating drug-acrolein adducts can trigger new health risks [181]. The biological fate of adducts formed due to drug-acrolein or drug-acrolein–protein interactions is highly complicated and not always fully understood. Attention should be paid to the appropriate acrolein scavenger doses to obtain a balanced benefit–risk effect [182]. Acrolein scavengers with proven effectiveness in in vitro/in vivo studies may, in the future, be effective and safe ways to reduce acrolein toxicity and related disorders. However, their bioavailability, appropriate concentrations and import into acrolein-damaged cells and tissues would require extensive research. A better understanding of the effect of acrolein on respiratory cells may shed new light on the mechanisms underlying the pathogenesis of, i.a., COPD, and pave the way for new ways to treat and prevent the disease [13].

## 9. Conclusions

This review was intended to summarize the biological molecular mechanisms of acrolein activity in the pathogenesis of lung diseases and cancers. Acrolein, one of the main pathogenic components of cigarette smoke, is considered one of the most harmful substances for tobacco users—both traditional smokers and smokers of e-cigarettes. Numerous in vitro and in vivo studies, through a logical sequence of studies and acrolein–biological effect relationships, provide indisputable evidence for the critical role of acrolein from cigarette smoke in the pathogenesis of COPD and lung cancer. Acrolein operates at various structural levels of cells, starting with cell organelles such as membranes, mitochondria, and ER, through to cells of the immune system, and ending with DNA. Exposure to acrolein through inhalation of cigarette smoke causes, among others: the inactivation of a number of proteins through direct interaction with acrolein, a cycle of oxidative damage caused by both inactivation of defense systems and direct generation of acrolein-dependent ROS, cell necrosis and chronic inflammation through regulation of activity/expression of crucial immune modulators, which over time causes the remodeling and degeneration of the respiratory tissue structure, ultimately manifesting as COPD. In addition, acrolein, as a powerful genotoxic agent, is responsible for most of the DNA damage observed in the cells of the respiratory system and lungs of smokers.

Undoubtedly, the biological activity of acrolein underlies the pathology and development of lung diseases and damage, including COPD and respiratory cancers. Several questions regarding the complex mechanisms of acrolein’s molecular activity are yet to be answered. It is necessary to establish detailed studies of the acrolein-dependent time sequence of molecular events and the inter-relationship, crosstalk and relative contribution of the different mechanisms of acrolein toxicity in the pathogenesis/progression of lung diseases. Undoubtedly, the development of small and large mammalian models of exogenous and endogenous acrolein exposures, both acute and chronic, may be helpful in the study of the detailed mechanisms of acrolein toxicity in humans.

## Figures and Tables

**Figure 1 cells-12-00879-f001:**
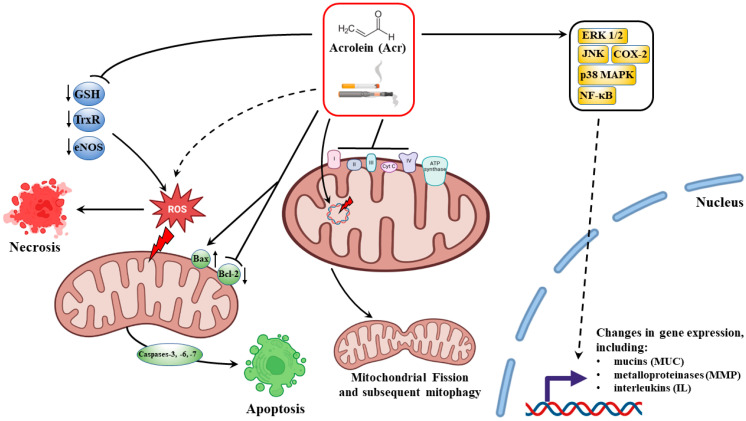
Molecular mechanisms of the cytotoxic effect of acrolein (Acr) in smokers are realized on several levels of cell functioning. Acrolein, through direct interaction with antioxidant enzymes (primarily GSH), disrupts redox homeostasis and contributes to the generation of ROS and the necrosis/apoptosis dependent on them. The negative effect of acrolein on mitochondrial biogenesis is also of great importance to damage to lung and respiratory system cells, i.e., inhibition of the activity of respiratory chain complexes and change of metabolism to glycolysis, mtDNA damage, mitochondrial fission, which is followed by autophagy/mitophagy. Due to the effect of acrolein on cell signaling pathways and direct interaction with numerous transcription factors (the main target of NF-kB), the aldehyde has strong immunomodulatory properties (regulation of pro-inflammatory cytokines). In addition, through the control of mucin (MUC) and metalloproteinase (MMP) gene expression, acrolein is involved in mucus hypersecretion and tissue modification and degradation in lung diseases.

**Figure 2 cells-12-00879-f002:**
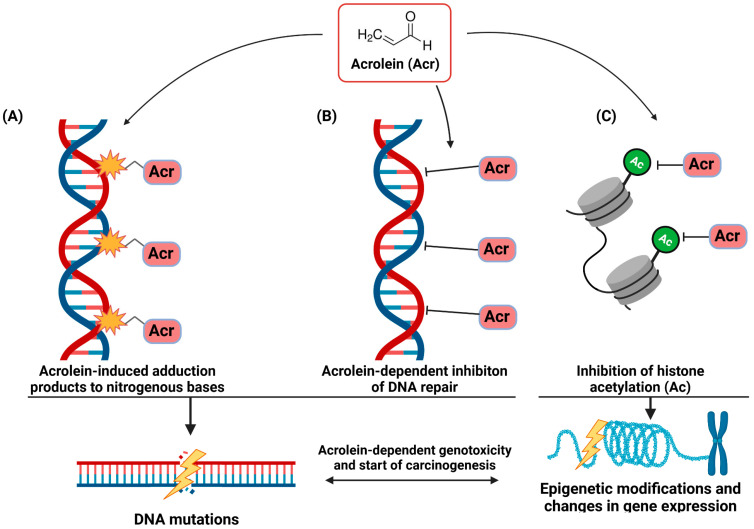
The genotoxic properties of acrolein (Acr) seem crucial in inducing cancer mutations in smokers. Acrolein interacting directly with nucleotide bases forms Acr-nucleotide base adducts and thus leads to their chemical modification (**A**). Effective acrolein-dependent weakening of the activity/inhibition of DNA repair enzymes amplifies the resulting DNA damage and mutations in cells (**B**). Additionally, acrolein affects the epigenetic mechanisms of gene expression regulation by inhibiting histone H4 acetylation (**C**).

## Data Availability

Data sharing not applicable.

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
