# Peer review of "The Tobacco Smoke Component, Acrolein, as a Major Culprit in Lung Diseases and Respiratory Cancers: Molecular Mechanisms of Acrolein Cytotoxic Activity"

_cells, 2023, doi:10.3390/cells12060879_

Round 1
Reviewer 1 Report
In line 161-162 you have mentioned "Exposure to e-cigarette smoke causes pulmonary and cardiovascular injury, heart rate and blood, endothelial dysfunction, and even DNA mutations leading to cancer". I recommend you to clarify what exactly happens to blood. It's a little bit vague.
Reviewer 2 Report
The manuscript ‘The tobacco smoke component, acrolein, as a major culprit in 1 lung diseases and respiratory cancers: molecular mechanisms 2 of acrolein cytotoxic activity’ was written well but the author will check the minor grammar mistakes and focus on the English part as well. Apart from this, the author should do further modifications as given in the comments to improve the current status of this manuscript.
Minor comments:
1. Add the reference at line no 110
2. Add the reference at line no 222 ,224 and 361
3. In table 1 why it was written as table 8, confirm it and delete it.
4. An author make sure that the mark increase and decrease should be before a word, not an after.
Major comments:
1. Author should include a short paragraph on other carcinogens present in tobacco and their risk of human health.
2. Acrolein is also responsible for causing heart diseases, COPD which include emphysema and chronic bronchitis. So along with lung cancer also include the details that how it causes other diseases.
3. Write a future aspects and then write a separate paragraph regarding conclusion.
Reviewer 3 Report
cells-2250966-peer-review-v1 reports
The manuscript with the title of “The tobacco smoke component, acrolein, as a major culprit in lung diseases and respiratory cancers: molecular mechanisms of acrolein cytotoxic activity” by Pawel Hikisz and Damian Jacenik,aimed to address current states of knowledge on the pathological molecular mechanisms of acrolein in the induction, course and development of lung diseases and respiratory cancers in smokers.
Some concerns should be well addressed on this manuscript as follows,
1) The title was not that clear but could be improved. When I read about the manuscript, I thought that the authors not only mainly wanted to discuss several molecular mechanisms of acrolein cytotoxicity.
2) The abstract was long. Words only from Line 19 to Line 22, mentioned the molecular mechanisms of acrolein cytotoxic effects.
3) Table 1 was so long and large. Why Table 8 was shown in Table 1? Was there something wrong?
4) I guess later on the readers may notice that big difference between the direct acrolein exposure and the cigarette smoke extract. E-cigarette exposure may also be different. In the Table 1, “Intranasal instillation of acrolein” and “Acrolein inhalation” were used in in vivo studies. However, the similarity and distinct comparison of all listed methods were not well explained.
5) Line 418, the authors wrote that “GSH is not the only enzyme involved in redox signalling, which is a direct target for acrolein.” In fact, GSH no matter its oxidized form or reduced form, is not an enzyme.
6) Acrolein can react with cellular proteins and enzymes and genes, but the site specificity of acrolein was not summaried and discussed in this manuscript.
7) And acrolein probably has already modified the ACE-2 on the lung cells. Subsequent meaning of this type of modification may be this alters the entry of coronaviruses like omicron. I do not know, but if this is to certain extend true, would this be interesting to be discussed? Acrolein is hazardous, but we still are talking about the biological molecular effects of acrolein in human lung health and diseases, from another angle and view.
8) If the authors wrote thioredoxins as “Trxs”, peroxiredoxins should be written as “Prxs” but not “prxs”, in this manuscript.
9) Typos in the manuscript should be revised.
10) Figure 2 should be improved to be better in quality.
11) English writing could be improved.
Overall, the submitted manuscript was tried to be focused, but the presentations were not properly adequate for immediately publishing on the journal Cells. At the present, it is not suitable for publishing directly on the journal Cells in its current version. Herein, I may recommend a major revision for this manuscript.
Round 2
Reviewer 2 Report
Authors have revised the work.
Reviewer 3 Report
The manuscript with ID of cells-2250966 has been revised and nearly all concerned issues have been responded nicely. I am satisfied with the answers from authors and glad to see the improvements in the table and figure of the manuscript.
I may suggest this present manuscript to be accepted on Cells.